



# General Circulation Models evaluation at different time scales over tropical region using ESA-CCI satellite data records: a case study of water vapour and cloud cover

Cedric Gacial Ngoungue Langue[1,2], Helene Brogniez[1,2], and Philippe Naveau[2]

[1]Laboratoire Atmosphères, Milieux, Observations Spatiales (LATMOS) - UMR 8190 CNRS/Sorbonne Université/UVSQ, 78280 Guyancourt, France.

[2]Laboratoire des Sciences du Climat et de l'Environnement, CEA Saclay l'Orme des Merisiers, UMR 8212 CEA-CNRS-UVSQ, Université Paris-Saclay & IPSL, 91191 Gif-sur-Yvette, France.

**Correspondence:** Ngoungue Langue C. G. (cedric-gacial.ngoungue-langue@latmos.ipsl.fr)

**Abstract.** Water vapour and cloud cover are two essential components of the earth's atmosphere. General circulation models (GCM) are used to study the long term evolution of the Earth's climate over past and future periods. The present work consists of assessing the representation of total column water vapor (TCWV) and total cloud cover (TCC) in the Atmospheric Model Intercomparison Project Phase 6 (AMIP6), the ERA5 reanalysis from the European Centre for Medium-Range Weather Fore-

casts (ECMWF), and satellite data records from the European Space Agency Climate Change Initiative (ESA-CCI). ESA-CCI is used as the reference for the common observation period with AMIP6, spanning from July 2003 to December 2014, to calibrate the framework. For the period prior to the observational period, from January 1981 to June 2003, ERA5 serves as the reference. This study is carried out over the tropical region which has been splitted in two sub-regions : the tropical oceans and tropical lands. The assessment of TCWV and TCC at different time-frequency is performed using a mathematical tool

called "multi-resolution analysis" (MRA). By applying the MRA decomposition, we found that the AMIP6 models produce consistent evolution of TCWV and TCC at seasonal to interannual scales (from 2 months to 5.6 years) in the tropical region, even if the representation of the amplitude of TCC remains sometimes challenging. The evaluation of ESA-CCI TCWV and TCC variability in AMIP6 models reveals that the models do not perform well at daily and subseasonal scales. At seasonal to interannual scales, the models reproduce more accurate variability of TCWV and TCC with respect to ESA-CCI. However,

AMIP6 models do not capture the trend in the evolution of ESA-CCI TCWV and TCC. The co-variations between TCWV and TCC were analyzed in the Niño3.4 region, revealing a significant positive correlation at the subseasonal scale, with a value of 0.7 for ESA-CCI and 0.3 for AMIP6. At seasonal to annual scales, we found a strong positive correlation between TCWV and TCC, with the exception of the CanESM5 and IPSL models, which showed a negative but significant correlation around -0.5.

## 1 Introduction

Climate modellers use reliable observation datasets to initialize and constrain the General Circulation Models (GCM). Reliable observations are necessary for a proper evaluation of the models. To support this need, the Global Climate Observing System (GCOS) highlights the importance of consistent, long-term records of key atmospheric variables critical to the United Nations



Framework Convention on Climate Change (UNFCCC), known as "Essential Climate Variables" (ECVs) (Hollmann et al., 2013). In response to the GCOS, the European Space Agency launched the Climate Change Initiative (ESA-CCI) project in
2008 for the purpose to produce coherent satellite data records of ECVs with high spatial and temporal resolution. These data are valuable for detecting processes at local scales and thus improving the skills of the models. Out of the 26 ECVs provided by the ESA-CCI projects, this study focuses on water vapor and cloud cover.

Water vapour and clouds are two essential components of the Earth's atmosphere. Water vapour plays an important role in weather patterns, climate dynamics and the water cycle (Held and Soden, 2000). It is the most important natural greenhouse
gas in the atmosphere with strong positive feedbacks (Sherwood et al., 2010; Colman, 2003; Colman and Soden, 2021). Water vapour accounts for about $67\%$ of greenhouse effect while $CO_2$ contribution is about $24\%$, $O_3$ $6\%$ and the $3\%$ for the other greenhouse gases ($CH_4, N_2O$) (Trenberth, 2022). Futhermore, it has been widely demonstrated by numerous studies that water vapor amplifies the warming induced by anthropogenic forcings [e.g. Manabe and Wetherald (1967); Parker et al. (2007); Dai (2006); Allan and Soden (2008); Mieruch et al. (2014); Zhang et al. (2013); Colman and Soden (2021)]. Indeed, as the Earth
warms, the concentration of water vapour in the lower troposphere increases with temperature, increasing the evaporation rate and, consequently, the amount of atmospheric water vapour, thus contributing to atmospheric warming. Clouds are also important for global climate since they have a strong impact on solar and terrestrial radiation as well as on the formation of precipitations [e.g. Raval and Ramanathan (1989); Allen and Ingram (2002); Stephens (2005); Klein and Hall (2015)]. They influence the Earth's energy balance by reflecting sunlight (albedo effect) and trapping infrared radiation (greenhouse effect)
(Stevens and Bony, 2013). A slight change in cloud amount or a shift in the vertical distribution of clouds might have a considerable impact on the Earth's energy budget (Bony et al., 2015). In present day climate, on average, clouds have a cooling effect (Trenberth et al., 2015); however, cloud feedbacks remain a major source of uncertainty in climate prediction (Wielicki et al., 1995; Stephens, 2005; Forster et al., 2021).

The tropical region located between $30°N$ and $30°S$ latitudes around the equator, is pivotal in the Earth's climate system
due to its role in energy balance, atmospheric circulation and the water cycle (Trenberth et al., 2009). The excess of incoming solar radiation in the region is redistributed in the mid- and high- latitudes by Hadley and Ferrel cells, which tend to reinforce the global atmospheric circulation (Trenberth et al., 2009; Hastenrath, 2012). The tropics are a major source of atmospheric water vapour, and tropical clouds play an essential role in the Earth's radiation (Hastenrath, 2012). Given the importance of water vapour and clouds in the Earth's climate system, many studies evaluating their representation in GCM have been carried
out around the world and in the tropics. In the following "CMIP" will refer to coupled ocean-atmosphere models and "AMIP" will refer to atmospheric models forced by observed Sea-Surface Temperature (SST).

Allan et al. (2022) assessed the changes in water vapor from 1979-2020 in CMIP6 climate models using satellite data, ground-based observations and ERA5 reanalysis. They found a trend in the evolution of integrated column water vapor in observations and CMIP6 atmosphere coupled only simulations of about 1%/decade during the period 1988-2014. However,
historical CMIP6 simulations overestimate water vapor trends during this period by up to a factor of two compared to AMIP simulations. This is partly documented in previous studies [e.g. Kosaka and Xie (2013); Mitchell et al. (2020)] showing that





internal climate variability suppressed observed warming in this period. Vignesh et al. (2020) investigated the evolution of cloud fraction in CMIP6 models using the CALIPSO-CLOUDSAT satellite data over the period going from July 2006 to February 2011. The results showed an overestimation of around 3% of the cloud fraction in CMIP6 ensemble mean compared

to the observations in the tropics. In a follow-up to previous studies, He et al. (2022) carried out an evaluation of total column water vapor (TCWV) in a subset of AMIP6 models using ESA-CCI observations in the tropics over the period July 2003 to December 2014. Their methodology relied on the analysis of the seasonal distribution of water vapor (Probability density functions) in AMIP6, ERA5 reanalysis and ESA-CCI looking at tropical land and oceans independently. They found large differences in the distribution of TCWV over tropical land between CMIP6 and ESA-CCI. On the order hand, over tropical

oceans, AMIP6 models show very similar patterns in the distribution of TCWV to those observed in ESA-CCI.

All of these studies have assessed the representation of water vapor or cloud fraction in CMIP6 and AMIP6 models using satellite data, and while some processes are correctly reproduced, the others are not. Identifying the processes that are not well captured by the models and their corresponding temporal frequency, will be crucial for improving their representation and, thus, enhancing the models' performance. Since geophysical signals are made up of a mixture of patterns with different

frequencies of variability, an accurate way of evaluating GCMs is to analyse their behaviour at each frequency. This type of analyses is commonly used in signal processing to extract some parts of a signal at a given frequency or removing noise. Some studies such as Oh et al. (2003) applies the multi-resolution analysis on solar irradiance in order to separate the high-frequency components to the lower-frequency ones. In the present study, we applied the multi-resolution analysis to TCWV and total cloud cover (TCC) in order to separate the various components, enabling us to evaluate the AMIP6 models on different time

scales.

The remainder of the paper is organised as follows: in section 2, we present the data and the methodology adopted to carry out this study. Section 3 contains the main results of this work, and Section 4 provides the conclusions and future perspectives.

## 2   Data and Methods

### 2.1   Data

We use daily TCWV and Total Cloud Cover (TCC) from the AMIP6, the ESA-CCI and the fifth generation of the European Centre for Medium-Range Weather Forecasts reanalysis (ERA5), averaged over the tropical region.

#### 2.1.1   ESA-CCI

ESA-CCI is an international programme which aims to develop a suite of satellite data records of ECVs (Hollmann et al., 2013). These data records provide a major contribution to the IPCC assessment of the state of the climate and underpin the

climate value information chain (monitoring, modelling, climate services) that enables policies to be developed, decisions and effective actions to be taken. The ESA-CCI programme developed different projects associated to each of the 26 ECVs. Among those projects, we are interested in the ESA-CCI Water vapour (WV_cci) and ESA-CCI Cloud cover (Cloud_cci).





The WV_cci TCWV data record combine microwave observations (SSMIS, SSM/I, AMSR-E and TMI) over the ice-free ocean (Fennig et al., 2017, 2020), and near-infrared imagers (MERIS, MODIS-Terra and OLCI) over land, coastal ocean and

sea-ice (Lindstrot et al., 2012; Diedrich et al., 2015; Preusker et al., 2021). Table.1 (**i**) shows the characteristics of instruments used to build the TCWV data record. The TCWV is defined as the vertically integrated water vapour over the full column with units of kg/m$^2$. WV_cci TCWV data record are available from July 2002 to December 2017 with a spatial resolution of $0.05°$ x $0.05°$.

The Cloud_cci data records used in this study is obtained from the measurements made by the Advanced Very High Resolu-

tion Radiometer (AVHRR) on a series of satellite platforms (NOAA-16, NOAA-18, NOAA-19) (Stengel et al., 2020). Table.1 (**ii**) shows the characteristics of instruments used to retrieve the TCC data. The Cloud-cci data record provide many cloud properties (cloud mask/fraction, cloud phase, cloud top pressure/height/temperature, cloud optical thickness, cloud effective radius and cloud liquid/ice water path). These cloud properties are obtained using the Cloud_cci simulator (Eliasson et al., 2019), but the cloud characteristics available in the AMIP6 models do not permit a comparative analysis to be conducted. Therefore, in

this work, we use only the cloud cover for the analysis of the representation of clouds in the models. In order to obtain the cloud mask, we averaged the ascendant and descendant cloud mask components that are available at $0.05°$x $0.05°$ and then upscale, by averaging to reach $0.5°$x $0.5°$. We upscale the resolution of TCC in Cloud_cci to match the resolution of TCC in AMIP6 models. The Cloud_cci data record are available from April 2001 to December 2016.

### 2.1.2   AMIP6 models

The AMIP6 models are GCM that provide long-term trends of climate variables and are used to study different climate scenarios. AMIP6 simulations are only forced by SST and sea-ice concentrations from observations (Ackerley et al., 2018). This model configuration enables scientists to focus on the atmospheric model without the added complexity of ocean-atmosphere feedbacks in the climate system (Gates et al., 1999). Following the study of He et al. (2022), we selected the same seven GCM from all the models involved in the AMIP6 experiment based on the availability of the variables of interest (TCWV and TCC)

at daily time scale. A more detailed description of the seven AMIP6 models is provided in [Tab.2]. The AMIP simulations used are available from January 1950 to December 2014.

The WV_cci TCWV are limited to clear-sky conditions over land. In order to apply a cloud screen in AMIP6 models, He et al. (2022) carried out a series of analyses using different cloud mask threshold values at different pressure levels. They showed that by applying a cloud mask threshold of 50% (i.e. retaining only the pixel with a cloud mask of less than 50%),

AMIP6 produces consistent patterns with respect to WV_cci and ERA5. We have followed this recommendation of He et al. (2022) to extract AMIP6 TCWV grid points over the land area.

The cloud cover is provided in AMIP6 models at each pressure levels. In order to be consistent with the Cloud_cci, the TCC on all vertical levels at each grid point in the AMIP6 models is determined by calculating the maximum cloud cover overlap (Tian and Curry, 1989; Oreopoulos and Khairoutdinov, 2003). This approach consists of considering, at a given grid point, the

vertical level where cloud cover is at its maximum as the total cloud cover over all the vertical levels.





Regarding the differences in the instruments used to retrieve WV_cci TCWV, the analyses are carried out separately over land and ocean. The distinction is done using the land-sea masks available for the models (AMIP6 and ERA5). As in (He et al., 2022), the land region is defined as an area with at least a lsm>=50% and the ocean region as an area with a lsm<50%.

### 2.1.3 ERA5 reanalysis

ERA5 reanalysis provides hourly estimates of various climate variables for the entire globe using 137 hybrid sigma levels up to 80 km above the surface (Hersbach et al., 2020). TCWV and TCC are retrieved directly from ERA5 database and are available from January 1980 to July 2024. The spatial resolution of ERA5 is 0.25°x 0.25°. To be coherent with AMIP6 and ESA-CCI datasets, we averaged hourly TCWV and TCC into daily data and considered separately for land and oceanic regions. As with CMIP6, some filters are applied to TCWV in ERA5 (see He et al. (2022) for more details). Over land, TCWV data with a

total cloud cover of less than 95% and a total liquid water column in clouds of less than $0.005 kg.m^{-2}$ are selected (Sohn and Bennartz, 2008). Over the oceans, TCWV data with total precipitation of less than $0.001 kg.m^{-2}.s^{-2}$ are selected.

ESA-CCI, ERA5 and AMIP6 data are not available for the same time periods. To ensure a proper evaluation of the models, we distinguish between two distinct periods. Figure 1 illustrates these two periods : the observed period from July 2003 to September 2014, where CMIP6, ERA5 and ESA-CCI data record are all available, and the pre-ESA period from January 1981

to June 2003, during which ESA-CCI data record are unavailable. For the evaluation of AMIP6 and ERA5, ESA-CCI data record are used as the reference during the observed period, while ERA5 serves as the reference for the pre-ESA period. The assessment of the representation of WV_cci TCWV and Cloud_cci TCC in AMIP6 is conducted over the observed period. Based on the results from this period, and assuming climate stationarity, we can infer the behavior of AMIP6 models relative to ESA-CCI across different time scales during the pre-ESA period as well as in the future.

## 2.2 Methods

### 2.2.1 The Multi-resolution analysis framework

Real world signals consist of a mixture of different components. Quite often, you are interested only in a subset of these components. There are many techniques that can be used to address this issue and the most popular are : Fast Fourier Transform (FFT), Continuous Wavelet Transform (CWT), Discrete Wavelet Transform (DWT) which includes Multi-resolution Analy-

sis (MRA) and Wavelet Packets Transform (WPT). FFT converts a time-domain signal only into its frequency components (Cochran et al., 1967; Nussbaumer and Nussbaumer, 1982; Rao et al., 2010). FFT excels at frequency resolution but lacks the ability to provide detail information about the changes in frequency over time. This limitation arises because FFT assumes that the input signal is periodic and stationary (Cerna and Harvey, 2000). In comparison with the FFT, CWT and DWT provide the decomposition of a signal in time-frequency domain but they have some specific characteristics. CWT offers continuous

time and frequency resolution, it requires high computational ressources (Sadowsky, 1996). DWT, on the other hand, provides a multi-resolution analysis, allowing the decomposition of a signal into different frequency components with varying time





resolutions (Gómez et al., 2016). DWT especially MRA, has been widely used in signal processing in order to extract features in a signal at a specific resolution. However, they are not used as much in atmospheric sciences for analysing climate data.

In this study, we applied the MRA to assess the temporal variability of water vapour and cloud cover signals at different frequencies. MRA is a mathematical tool which consists of decomposing an original signal into its subcomponents at different resolutions (Oh et al., 2003; He and Guan, 2013; Lin and Franzke, 2015; Gómez et al., 2016; Behzadpour et al., 2019). MRA is built on Discrete Wavelet Transform (DWT) which allows the decomposition of a discrete signal (temporal or spatial) into a low frequency component (or approximation of the initial signal usually interpreted as the trend) and a high frequency component (or noise, usually regarded as fluctuations). This decomposition can be repeated for the low and high frequency components at
different resolutions.

The decomposition in MRA of a climatic signal gives two important components: the long-term evolution in the signal ($\mathbf{S_n}$ which corresponds to the wavelet smooth vector) and the variability of the signal at different temporal resolutions ($\mathbf{D_j}$ which corresponds to the wavelet detail vector) [Fig.1]. Therefore, the signal $\mathbf{S}$ can be rewritten as follows:

The decomposition in MRA of a climatic signal $\mathbf{S}$ gives two important components: its long-term evolution and different
variability contributions explained by the following equation :

$$\mathbf{S} = \mathbf{S_n} + \sum_{\mathbf{i=1}}^{\mathbf{n}} \mathbf{D_i} \tag{1}$$

where $\mathbf{n}$ is the maximum level of decomposition, In Equation (1), $\mathbf{S_n}$ corresponds to the so-called smooth vector, see e.g. Equation (2) in (Oh et al., 2003) and each term $\mathbf{D_i}$ corresponds to the so-called detailed signal with respect to level i (ie., the variability at level i) [Fig.2]. The expressions of $\mathbf{S_n}$ and $\mathbf{D_i}$ are given in appendix.

To apply the MRA decomposition on a temporal signal, its length must be dyadic (a multiple of power of 2). In practice, this simply means that the number of days between July 2003 and December 2014 is equal to 4202 and be expressed as $4096 = 2^{12}$ in our application and, in this case, we have 12 levels of resolution with level 1 indicating high resolution (daily scale) and level 12 corresponding to low resolution (low climatology frequency). Figure 2 illustrates the MRA decomposition of the TCWV, showing that the signal components exhibit a high frequency of variability at short time scales (from 1 to 4 days) and a low
frequency of variability at long time scales (from 1 to 6 years).

### 2.2.2 Evaluation of the WV_cci TCWV and Cloud_cci TCC variability in AMIP6 simulations at different frequencies

The scope of this study is to evaluate the times series of water vapour and clouds for some AMIP6 models at all frequencies starting at the daily scale, with respect to ESA-CCI data records (TCWV and TCC). To do so, we have implemented the following method, illustrated by [Fig.3]:

1. The ESA time serie is decomposed by the MRA which gives 12 detail vectors $\mathbf{D_i^{ESA}}$ and one smooth vector $\mathbf{S_{12}^{ESA}}$ representing the long term evolution of the signal.





2. Similarly, a given $GCM_j$ is decomposed with the MRA into 12 detail vectors $\mathbf{D_i^{GCM_j}}$ and one smooth vector $\mathbf{S_{12}^{GCM_j}}$.

3. A proxy of the ESA time serie, called "$\mathbf{ESA\_GCM_j\_D_i}$" is reconstructed by replacing, at the level i, the detail vector $\mathbf{D_i^{ESA}}$ by the corresponding $\mathbf{D_i^{GCM_j}}$, and all other components coming from the ESA decomposition are kept irreplaceable. The same technique is applied for different levels and the smooth vector $S_n$.

4. The "$\mathbf{ESA\_GCM_j\_D_i}$" is then compared to the original ESA signal by computing the root mean square (RMSE) and correlation coefficient.

All these steps are followed for all the AMIP6 models separately for TCWV and TCC. This recombination technique makes it possible to evaluate each of the frequency components ($D_i$ and $S_n$) of the initial signal and to identify the time scales over which the AMIP6 models reproduce properly the evolution of the ESA-CCI time series.

## 3 Results

### 3.1 Evolution of water vapour and cloud cover at different time scales

#### 3.1.1 First insight of the co-variations between TCWV, TCC and SST

The evolution of the TCWV, TCC and SST anomalies over the tropical oceans during the observed period is analyzed in Figure 4. The anomalies are computed by removing the climatology in the evolution of signals. We can notice some similarities in the evolution of the TCWV and SST in the region. During some specific el niño events such as the 2006/2007, 2009/2010 and 2014/2015, we observed an increase in water vapour content [Fig4 (a,c)]. The analysis of the correlation between the evolution of the SST and TCWV anomalies shows significant values of around 0.5 over the tropical oceans [Table 3]. Most of the AMIP6 models represent this link between the evolution of the TCWV and SST. This suggests that the el niño phenomenon during its activity has a greater influence on the water vapour content over the tropical oceans than over the tropical lands. This is not surprising knowing that El niño phenomena is characterized by an abnormal warming of surface waters, which increases the evaporation rate, leading to higher water vapor content. Regarding the evolution of TCC and SST anomalies, we did not find any major links and the correlation values computed are very low (not shown).

#### 3.1.2 MRA decomposition of TCWV and TCC times series

Using the MRA approach, we have assessed the temporal evolution of TCWV and TCC at four major time scales of great importance for climate and weather: daily (1-8 days from $D_1$ to $D_4$), subseasonal (2 weeks - 1 month from $D_5$ to $D_6$), seasonal to annual (2 months - 1.4 year from $D_7$ to $D_{10}$) and annual to decadal (3 - 6 years from $D_{11}$ to $D_{12}$). For each time scale, we averaged the signal components overall the $D_i$ frequencies belonging to the corresponding time scale. In this section, we do not present the results obtained at daily scale because their analysis is not relevant in view of the high frequency of variability observed in the evolution of the TCWV and TCC signals and the number of atmospheric processes that come into





play at this time scale in the tropical region (e.g. diurnal cycle, Mesoscale Convective System). As mentionned previously, the analyses are conducted over tropical oceans and lands independently.

Figure 5 shows the MRA decomposition of TCWV and TCC from subseasonal to interannual scales. At subseasonal scale, we observed large differences in the amplitude and phases of the signals (TCWV, TCC) between the AMIP6 models and ESA-CCI [Fig.5 (a)]. At seasonal and interannual scales, the evolution of the TCWV and TCC becomes smoother than that observed at daily and subseasonal scales. The AMIP6 models produce a coherent evolution of TCWV with respect to ESA-CCI, while the representation of the amplitude of TCC remains challenging [Fig.5 (b,c)]. The evolution of TCWV and TCC over tropical land is similar to the results found over the oceans, with a few exceptions on annual to decadal scales, where the AMIP6 models do not reproduce a consistent evolution compared to ESA-CCI (see [Fig.S1 (c)] in supplement material). The MRA decomposition of ERA5 signals (TCWV, TCC) shows that ERA5 produces an evolution closer to ESA-CCI over tropical ocean than tropical land. This confirms that ERA5 presents a significant bias over tropical land, which it is important to be aware of before using it as a reference for model evaluation. This significant bias of ERA5 compared to ESA-CCI on tropical lands has also been highlighted by He et al. (2022).

### 3.1.3 Analysis of trends in the evolution of TCWV and TCC

Figure 6 shows the evolution of the trend in the different AMIP6 models, ESA-CCI and ERA5 over the tropical oceans. More generally, the amplitude of the trend of TCWV with respect to ESA-CCI is underestimated by most of the AMIP6 models (38 vs $35 kg/m^2$) over the tropical region, with the exception of the IPSL model, which shows an overestimation (38 vs $39 kg/m^2$) [Fig.6]. The MPI model produces an amplitude of the trend of TCWV close to ESA-CCI both over tropical land and ocean. The amplitude of the trend of TCC with respect to ESA-CCI is also underestimated in AMIP6 models with the exception of NCAR models (CESM2, CESM2-WACCM) which show a slight overestimation over tropical land (not shown).

We argue that the trend is not linear over the whole period, but at the end of the period, from 2010 to 2014, the trend appears to be linear. Therefore, we compute the linear trend in the evolution of TCWV and TCC over the 4 last years of the study period, using the Theil-Sen estimator (Akritas et al., 1995). It is a robust estimator that can be used to fit a line to samples of points in the plane by choosing the median of the slopes of all the lines passing through pairs of points. We recognize that this four-year trend does not represent a climatological trend; however, it is valuable for evaluating the evolution of TCWV and TCC in the AMIP6 models over this specific period. The results of the trend analysis over the tropical region are shown in [Tab.4]. ESA-CCI shows a major difference between trends in the evolution of TCWV over the tropical region. The TCWV trend over tropical lands is about seven times that over tropical oceans ($0.2 \pm 7.10^{-4}$ kg/m$^2$/yr and $0.03 \pm 10^{-4}$ kg/m$^2$/yr). ERA5 is able to capture the trend in the evolution of the TCWV over the tropical oceans. Most of the AMIP6 models do not capture this trend in the tropical region, although we have noticed a few exceptions with the MPI model over the tropical oceans, which shows a trend of $0.02 \pm 6.10^{-4}$ kg/m$^2$/year. As far as the evolution of the TCC is concerned, we did not find any significant trend in the tropical region.





## 3.2 Evaluation of the representation of TCWV and TCC variability at different time scales

### 3.2.1 Evaluation over the observed period

After assessing the evolution of TCWV and TCC in the AMIP6 and ERA5 models with respect to ESA-CCI, we are now interested in the representation of the variability of ESA-CCI signals in the AMIP6 models and ERA5 at different frequencies. This is done following the methodology described in section 2.3.2 using two statistic metrics : RMSE and Pearson correlation. Figure 7 shows the values of the two metrics with respect to ESA-CCI over the tropical oceans for TCWV and TCC. The AMIP6 models show a much lower correlation ( 0.91) associated with RMSE values of around  1.5 kg/m$^2$  and  1.5% at daily

and subseasonal scales for TCWV [Fig.7 (i)] and TCC [Fig.7 (ii)] respectively. This could result from a misrepresentation of synoptic processes in AMIP6 models. At seasonal to annual scales, the AMIP6 models show a much higher correlation with values ranging from (0.97-1)/(0.94-1) associated with low RMSE values of (0-0.5 kg/m$^2$)/(0.25-1%) for TCWV and TCC respectively. However, we noticed a specific behaviour of the CNRM and MPI models which show RMSE values around 1.5% for the TCC signal at 4.2 months and 1.4 year [Fig.7 (ii)]. AMIP6 models reproduce more accurately the variability of

ESA-CCI TCWV and TCC at seasonal to annual time scales. This suggests that they better represent large-scale processes. However the representation of the trend in the evolution of ESA-CCI TCWV and TCC remains challenging for the AMIP6 models. This result confirms the findings on the trends analysis in the previous section. Over tropical land, we obtained similar results to those over oceans for TCWV, but we noticed some differences with TCC. We found much higher RMSE values at seasonal to annaual scales than those obtained over tropical oceans, with one exception at 4.2 months (see [Fig.S2] in the

supplement material). This implies that the AMIP6 models reproduce the variability of the TCC over the ocean better than over land at seasonal scale. The analysis of the representation of ESA-CCI TCWV and TCC variability in ERA5 over tropical oceans reveals significant RMSE values of approximately  2kg/m$^2$ around  5.6 years, whereas the RMSE values are lower for the rest of the frequencies.

Based on the previous results showing the RMSE between the "**ESA_GCM$_j$_D$_i$**" and ESA-CCI at different frequencies,

we built a new AMIP6 estimator called "Best_AMIP6". The Best_AMIP6 is a blend of AMIP6 models with the lowest RMSE values relative to ESA-CCI at each decomposition frequency. The Best_AMIP6 is then evaluated with respect to ESA-CCI by computing Pearson correlation and a quantile analysis. Figure 8 **(i)** shows the correlation between the (Best_AMIP6 and ERA5) **(a)** and (Best_AMIP6 and ESA-CCI) **(b)** for TCWV over tropical oceans. The same applies for TCC in Figure 8 **(ii)**. In addition, Table 5 shows the correlation between AMIP6 and ESA-CCI, including the Best_CMIP6. We observed a slight

improvement in the correlation when using the Best_AMIP6 model for TCWV and TCC signals both over tropical land and ocean [Tab.5]. The quantile analysis shows that the Best_AMIP6 model did not perform well for the extreme values of TCC and TCWV. The Best_AMIP6 model shows higher correlation with respect to ESA-CCI for TCWV/TCC over tropical ocean (0.7/0.55) than land (0.63/0.28) [Fig.8 (b,d) and Fig.S3 (b,d) in the supplement material]. This behavior in the AMIP6 models can be explained by the use of AMIP simulations which are forced by the sea surface temperature leading to good correlation

over the ocean. AMIP6 models better represent the evolution of TCWV than that of TCC in the tropics. This may result from





the complexity of representing cloud microphysics and dynamics in large grid cells, as is the case with GCM (Tompkins, 2005; Siebesma and Seifert, 2020). The correlation between the Best_AMIP6 and ERA5/ESA-CCI is very close over tropical oceans [Fig.8]. This is consistent with our previous results showing that ERA5 is very close to ESA-CCI over the tropical oceans. Therefore, assuming that the evolution of the Earth's climate is stationary, ERA5 can be used as a reference to evaluate AMIP6 models over tropical oceans during historical and future periods for which satellite data are not available.

### 3.2.2 Evaluation over the pre-ESA period

The previous set of analyses was carried during the observed period. Assuming stationnarity in the evolution of TCWV and TCC, we could apply the results found on the reconstruction of ESA-CCI using AMIP6 models in order to derive the Best_AMIP6 over the historical period. This suggests that for each level of decomposition, the relationship found between the AMIP6 models and ESA-CCI in terms of RMSE and correlation over the observed period is preserved [Fig.7]. Figures 9 shows the evolution of TCWV and TCC over the pre-ESA period over tropical oceans. The Best_AMIP6 model reproduces a very close evolution of the TCWV compared to ERA5 on seasonal and annual scales, whereas with the TCC, we noticed some discrepancies between the Best_AMIP6 and ERA5. This is quite interesting knowing that ERA5 and ESA-CCI water vapour signal are very close over tropical ocean at the same scales over the observed period. More generally, the AMIP6 models reproduce similar evolution over pre-ESA and observed periods. In Table 6, we present the correlation between AMIP6 models and ERA5. The correlation between the evolution of TCWV in the Best_AMIP6 and ERA5 is not too different over the observed period (0.69) and the pre-ESA period (0.64). We can also notice that the Best_AMIP6 does not show the highest correlation compared to ERA5, which can be explained by the fact that it has been fitted using ESA-CCI data which show some differences with the ERA5 reanalysis. We found similar results over tropical lands (see Fig.S4 in supplement material).

### 3.3 Analysis of the co-variations between TCWV and TCC in the Nino 3.4 region

After assessing the evolution of the TCWV and TCC signals, we are now interested in their co-variations: how do they vary together? This analysis is very important given the strong dependence between the two variables, as clouds form from the condensation of water vapour. Our region of study, the tropical band is characterized by a strong convective activity. Investigating the co-variations of TCWV and TCC in this region is very difficult because of large- and local- scales processes coming into play. Following the work of Wagner et al. (2005), which highlighted the influence of El niño on cloud cover and the total column precipitable water, we reduced, especially for this analysis, the study region to the Nino3.4 region. The Nino3.4 region is located over the Pacific ocean at $[170°W − 120°W, 5°N − 5°S]$ (see Fig.S5 in the supplement material). The co-variations between TCWV and TCC are assessed by computing the correlation between the two variables at different frequencies of variability. The results are shown in Figure 10. We found a non significant correlation between TCWV and TCC at daily scale from 1-4 days in all the products (ESA-CCI, AMIP6 and ERA5). This is not surprising knowing that TCWV and TCC exhibit a high-frequency variability at daily scale, which makes it difficult to assess the co-variations between the two variables at this scale. At subseasonal scale, all the products show a positive and significant correlation in the evolution of TCWV and TCC at around 0.7 for ESA-CCI and 0.3 for CMIP6. We observed at seasonal to annual scale a strong positive correlation





between TCWV and TCC, with the exception of the CanESM5 and IPSL models, which showed a negative but significant
correlation around -0.5. At interannual scale, we have noticed a particular behaviour between ERA5 and ESA-CCI, while
ESA-CCI shows a strong positive correlation of 0.97 between TCWV and TCC, ERA5 shows a strong negative correlation of
-0.98, which indicate an opposite evolution between the two variables. This negative correlation found in ERA5 appears at the
same frequency as the large RMSE values identified when evaluating the representation of the variability of the TCWV and
the TCC of the ESA-CCI in ERA5. The behaviour of ERA5 at this specific frequency need to be investigated in more detail.
Most of the AMIP6 models show this strong positive correlation as observed in ESA-CCI, while the others (IPSL, MPI and
CESM2) show a negative correlation as is the case with ERA5.

## 4 Conclusions

In this work, we evaluated the representation of TCWV and TCC in AMIP6 simulations, ERA5, and ESA-CCI satellite data
records across different time scales, using MRA, a powerful tool for signal decomposition. The analyses were conducted
separately for tropical oceans and land from July 2003 to September 2014 for the observed period, and from January 1981 to
June 2003 for the pre-ESA period.

Firstly, we analysed the evolution of TCWV and TCC at different time scales in ERA5, AMIP6 and ESA-CCI. We found
that AMIP6 models produce coherent evolution of TCWV and TCC with respect to ESA-CCI at seasonal and interannual time
scales (from 2 months to 1.4 year). However, the AMIP6 models underestimate the amplitude of the trend in TCWV with
respect to ESA-CCI (38 versus 35 kg/m2) in the tropical region, with the exception of the IPSL model, which shows a slight
overestimation (38 versus 39 kg/m2). The MPI model produces an amplitude of the trend of TCWV close to ESA-CCI. We
found similar results with TCC, with an exception of the NCAR models (CESM2, CESM2-WACCM) which show a slight
overestimation over tropical land.

Secondly, we evaluated the representation of the ESA-CCI variability in the AMIP6 models at different time scales. We found
that the AMIP6 models show a much lower correlation ( 0.91) associated with RMSE values of around 1.5 kg/m$^2$ and 1.5% at
daily and subseasonal scales for TCWV. This could result from a misrepresentation of synoptic processes in AMIP6 models.
At seasonal to annual scales, the AMIP6 models show a much higher correlation with values ranging from (0.97-1)/(0.94-1)
associated with low RMSE values of (0-0.5 kg/m2)/(0.25-1%) for TCWV and TCC respectively. AMIP6 models reproduce
more accurately the variability of ESA-CCI TCWV and TCC at seasonal to annual time scales. This suggests that they better
represent large-scale processes. Nevertheless, the representation of the trend in the evolution of ESA-CCI TCWV and TCC
remains challenging for the AMIP6 models. We also attempted a reconstruction of ESA-CCI TCWV and TCC signals using
AMIP6 models over the observed period. The The Best_AMIP6 model shows higher correlation with respect to ESA-CCI for
TCWV/TCC over tropical ocean (0.7/0.55) than land (0.63/0.28). This behavior in the AMIP6 models can be explained by
the use of AMIP simulations which are forced by the sea surface temperature leading to good correlation over ocean. The
reconstruction method does not perform well over the extreme values.



Thirdly, we studied the co-variations between TCWV and TCC over a small region in the tropical ocean ("Niño 3.4" region) in all the products (ERA5, AMIP6 and ESA-CCI) during the observed period. We found at subseasonal scale, a positive and significant correlation in the evolution of TCWV and TCC of around 0.7 for ESA-CCI and 0.3 for CMIP6. At seasonal to annual scales, we found a strong positive correlation between TCWV and TCC, with the exception of the CanESM5 and IPSL

models, which showed a negative but significant correlation around -0.5. At interannual scale, we have noticed a particular behaviour between ERA5 and ESA-CCI, while ESA-CCI shows a strong positive correlation of 0.97 between TCWV and TCC, ERA5 shows a strong negative correlation of -0.98, which indicates an opposite evolution between the two variables.

In future work, it will be interesting to reproduce the same analyses on fully coupled atmosphere-ocean AMIP6 simulations. Targeting daily and subseasonal scales, which seem the most problematic scales at the end of the above study, should be

carried out in some selected areas in order to deepend the synoptic processes in models. The application of the MRA to other observational datasets (such as GNSS network) is also important to identify potential temporal scales with similarities and discrepancies, induced by the different sampling in the observing system.

*Acknowledgements.* The study was funded by ESA via WV_cci project. The combined microwave and near-infrared-imager-based product COMBI was initiated and funded by the WV_cci project, with contributions from Brockmann Consult, Spectral Earth, Deutscher Wetterdienst

and the EUMETSAT Satellite Climate Facility on Climate Monitoring (CM SAF). The combined MW and NIR product will be owned by EUMETSAT CM SAF and will be released by CM SAF in late 2021. This study benefited from the IPSL ESPRI (Ensemble de Services Pour la Recherche à l'IPSL) computing and data center (https://esgf-node.ipsl.upmc.fr/projects/esgf-ipsl/).

*Competing interests.* The contact author has declared that none of the authors has any competing interests





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



**Tables**

**Table 1.** Description of the instruments used in the ESA-CCI data retrieval for: **(i)** TCWV and **(ii)** TCC. LTDN stands for "local time descending node". [1] 300km swath over sea and 512km over land. [2] 1420km nadir swath and 750km backwards (dual) view swath. [3] 0.5km visible channel resolution and 1km infrared channel resolution.

| | | | Characteristics | | | |
|---|---|---|---|---|---|---|
| Instruments | Spectral domains | Region | Data description | Spatial resolution (km) | Time span | Reference |
| MERIS | NIR | land, coastal and sea ice | daytime, cloud-free | 1.2 | 2002-2012 | Fischer and Bennartz (1997) |
| MODIS | NIR | land, coastal and sea ice | daytime, cloud-free | 1 | 2011-2017 | Gao and Kaufman (2003) |
| OLCI | NIR | land, coastal and sea ice | daytime, cloud-free | 1.2 | 2016-2017 | Lindstrot et al. (2012) |
| HOAPS | MW | ocean | 6-hourly composites | 0.5° | 2002-2017 | Lindstrot et al. (2014) |

**(i)**

| | | | Characteristics | | |
|---|---|---|---|---|---|
| Instruments | LTDN | Swath (km) | Spatial resolution (km) | Time span | Reference |
| ATSR-2 ERS-2 | 10.30 | 300/512[1] | 1 | 1995-2008 | Poulsen et al. (2019) |
| AATSR Envisat | 10.00 | 512/512 | 1 | 03/ to 04/2002 | |
| SLSTR Sentinel-3a | 10.00 | 750/1420[2] | 0.5-1 [3] | since 2016 | |
| SLSTR Sentinel-3b | 10.00 | 750/1420[2] | 0.5-1 [3] | since 2018 | |

**(ii)**





**Table 2.** Description of the Data used in this work.

| Institution | Models | Horizontal resolution | Vertical resolution | Percentage of land data | Percentage of ocean data | Reference |
|---|---|---|---|---|---|---|
| CCCma | CanEMS5 | 2.81°*2.81° | 49 levels (1-1022 hPa) | 55.63 % | 99.89% | Swart et al. (2019) |
| CNRM-CERFACS | CNRM-CM6-1 | 1.41°*1.41° | 91 levels (0.1-1039 hPa) | 62.85 % | 99.86% | Voldoire et al. (2019) |
| | CNRM-ESM2-1 | | | 62.81% | | Séférian et al. (2019) |
| IPSL | IPSL-CM6A-LR | 1.25°*2.50° | 79 levels (0-1028 hPa) | 76.10% | 99.79% | Lurton et al. (2020) |
| MPI-M | MPI-ESM1-2-HR | 0.94°*0.94° | 95 levels (0-1055 hPa) | 69.90% | 99.98% | Müller et al. (2018) |
| NCAR | CESM2 | 0.94°*1.25° | 32 levels (4-993 hPa) | 47.14% | 99.97% | Danabasoglu et al. (2020) |
| | CESM2-WACCM | | 70 levels (0-993 hPa) | 46.14% | | Gettelman et al. (2019) |
| ESA CCI | TCWV-COMBI | 0.05°*0.05° | - | 43.73% | 99.82% | Same as in Table.1 (i) |
| | Cloud-CCI | | - | | | Stengel et al. (2017) |
| ECMWF | ERA5 | 0.5°*0.5° | - | 52.76% | 97.14% | Hersbach et al. (2020) |

**Table 3.** Correlation with a significance level of 99.9% between the SST and TCWV anomalies over the tropical oceans.

| Models | ESA | ERA5 | CanESM5 | CNRM-ESM2-1 | CNRM-CM6-1 | IPSL-CM6A-LR | MPI-ESM1-2-HR | CESM2-WACCM | CESM2 |
|---|---|---|---|---|---|---|---|---|---|
| Correlation | 0.5 | 0.45 | 0.49 | 0.52 | 0.44 | 0.54 | 0.46 | 0.52 | 0.59 |

**Figures**

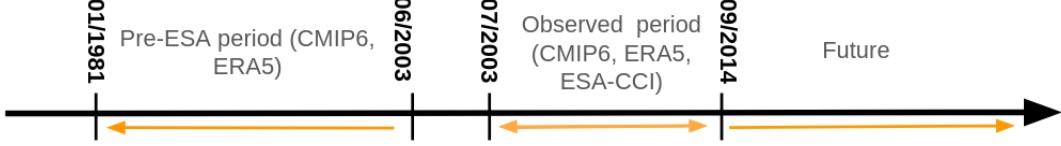

**Figure 1.** Representation of the study period from the pre-ESA period to the future.



**Table 4.** Trends per decade in the evolution of TCWV and TCC over the period 2003-2014.

| Models | TCWV | | TCC | |
|---|---|---|---|---|
| | *Land* | *Ocean* | *Land* | *Ocean* |
| *Models* | $trend(kg/m^2/year)$ | $trend(kg/m^2/year)$ | $trend(\%/year)$ | $trend(\%/year)$ |
| *ESA* | $0.2 \pm 7.10^{-4}$ | $0.03 \pm 10^{-4}$ | $-0.00 \pm 10^{-5}$ | $0.2 \pm 8.10^{-4}$ |
| *ERA5* | $-0.01 \pm 4.10^{-5}$ | $0.03 \pm 9.10^{-5}$ | $0.03 \pm 10^{-4}$ | $-0.00 \pm 9.10^{-6}$ |
| *CanESM5* | $0.01 \pm 5.10^{-5}$ | $0.01 \pm 5.10^{-5}$ | $-0.03 \pm 10^{-4}$ | $0.04 \pm 10^{-4}$ |
| *CNRM-ESM2-1* | $-0.04 \pm 10^{-4}$ | $-0.01 \pm 5.10^{-5}$ | $-0.1 \pm 3.10^{-4}$ | $-0.01 \pm 4.10^{-5}$ |
| *CNRM-CM6-1* | $-0.07 \pm 2.10^{-4}$ | $-0.00 \pm 2.10^{-6}$ | $-0.1 \pm 3.10^{-4}$ | $0.02 \pm 6.10^{-5}$ |
| *IPSL-CM6A-LR* | $-0.03 \pm 10^{-4}$ | $0.01 \pm 2.10^{-5}$ | $-0.03 \pm 10^{-4}$ | $-0.02 \pm 6.10^{-5}$ |
| *MPI-ESM1-2-HR* | $-0.01 \pm 5.10^{-5}$ | $0.02 \pm 6.10^{-4}$ | $-0.1 \pm 4.10^{-4}$ | $0.04 \pm 10^{-4}$ |
| *CESM2-WACCM* | $-0.06 \pm 2.10^{-4}$ | $-0.01 \pm 3.10^{-5}$ | $-0.08 \pm 3.10^{-4}$ | $0.04 \pm 10^{-4}$ |
| *CESM2* | $-0.04 \pm 10^{-4}$ | $0.00 \pm 5.10^{-6}$ | $-0.1 \pm 3.10^{-4}$ | $0.05 \pm 10^{-4}$ |

**Table 5.** Correlation between AMIP6 and ESA TCWV and TCC, including the "Best_CMIP6" over the observed period.

| Models | TCWV | | TCC | |
|---|---|---|---|---|
| | Land | Ocean | Land | Ocean |
| *Models* | *Correlation* | *Correlation* | *Correlation* | *Correlation* |
| *Best_CMIP6* | *0.63* | *0.7* | *0.28* | *0.55* |
| *CanESM5* | *0.56* | *0.58* | *0.21* | *0.49* |
| *CNRM-ESM2-1* | *0.52* | *0.67* | *0.19* | *0.07* |
| *CNRM-CM6-1* | *0.54* | *0.65* | *0.21* | *0.04* |
| *IPSL-CM6A-LR* | *0.51* | *0.67* | *0.09* | *0.53* |
| *MPI-ESM1-2-HR* | *0.63* | *0.56* | *0.2* | *0.34* |
| *CESM2-WACCM* | *0.57* | *0.51* | *0.1* | *0.22* |
| *CESM2* | *0.56* | *0.54* | *0.19* | *0.15* |



**Table 6.** Correlation between AMIP6 models and ERA5 TCWV and TCC, including the "Best_CMIP6" over the pre-ESA period.

| | TCWV | | TCC | |
|---|---|---|---|---|
| | *Land* | *Ocean* | *Land* | *Ocean* |
| *Models* | *Correlation* | *Correlation* | *Correlation* | *Correlation* |
| *Best_CMIP6* | *0.58* | *0.64* | *0.32* | *0.46* |
| *CanESM5* | *0.62* | *0.59* | *0.41* | *0.47* |
| *CNRM-ESM2-1* | *0.48* | *0.64* | *0.35* | *0.15* |
| *CNRM-CM6-1* | *0.49* | *0.63* | *0.32* | *0.1* |
| *IPSL-CM6A-LR* | *0.51* | *0.69* | *0.15* | *0.47* |
| *MPI-ESM1-2-HR* | *0.62* | *0.57* | *0.28* | *0.31* |
| *CESM2-WACCM* | *0.57* | *0.57* | *0.25* | *0.22* |
| *CESM2* | *0.56* | *0.53* | *0.25* | *0.22* |





**Figure 2.** MRA decomposition apply on a climatic signal "S" (total column water vapour) over tropical ocean. The decomposition is applied on 12 levels, we only show $D_1, D_3, D_5, D_{12}, S_{12}$. The top panel represents the evolution of the TCWV, the middle panels represent the components of the TCWV signal at different frequencies and the last panel shows the evolution of the trend.





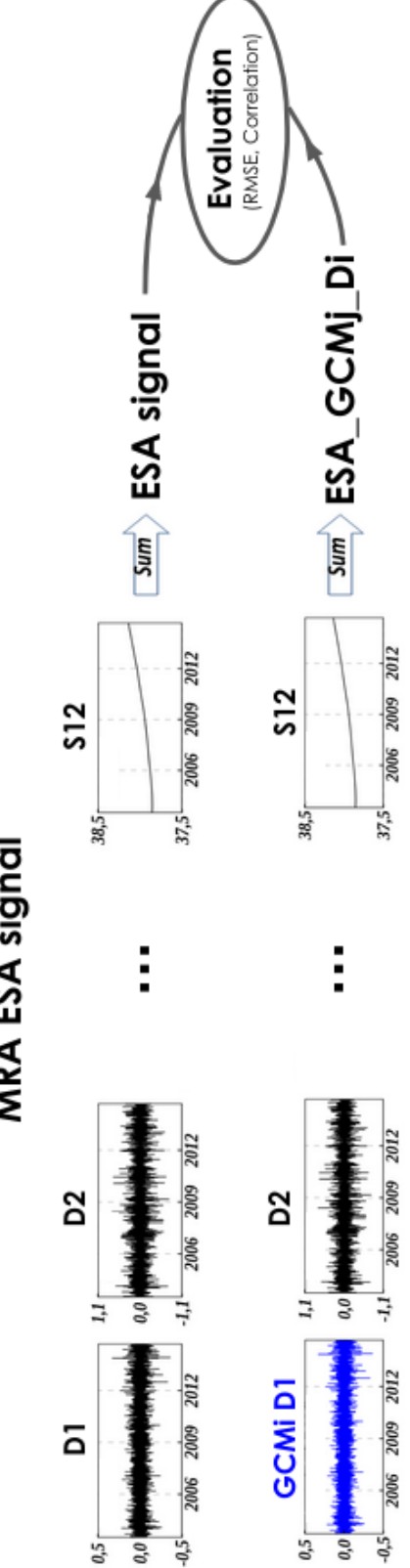

**Figure 3.** Schematic illustration of the methodology used to evaluate the representation of the variability of satellite observations in AMIP6 models at different frequencies.



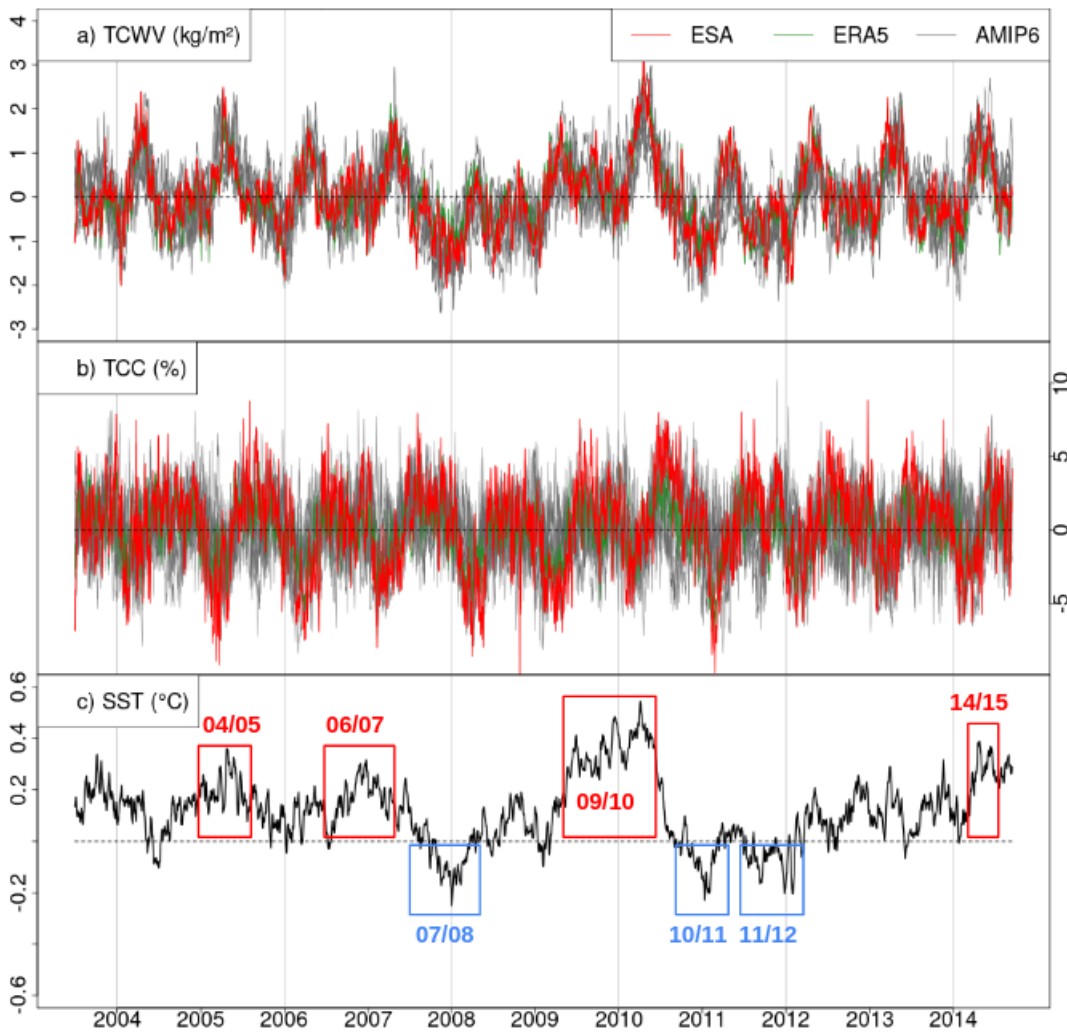

**Figure 4.** Evolution of the TCWV, TCC and SST anomalies over the tropical oceans during the observed period. The anomalies are computed by removing the climatology in the evolution of signals. The red, green and gray curves represent ESA-CCI, ERA5 and AMIP6 models respectively. The red and blue boxes represent El Niño and La Niña past events respectively. The Y-axis represent the anomalies of : **(a)** TCWV in $kg/m^2$ , **(b)** TCC in % and **(c)** SST in $°C$. The X-axis indicates the time.



**Figure 5.** MRA decomposition of TCWV and TCC signals over the tropical oceans during the observed period. The decomposition is applied on 12 levels of resolution and the results are grouped in three time frequencies : **(a)** subseasonal, **(b)** seasonal to annual and **(c)** annual to decadal. The red, green and gray curves represent ESA-CCI, ERA5 and CMIP6. The Y-axis represent TCWV in $kg/m^2$ and TCC in %. The X-axis indicates the time.







**Figure 6.** Evolution of the trends in the TCWV and TCC signals over the tropical oceans during the observed period. The different colored curves represent the AMIP6 models, ERA5 and ESA-CCI. The Y-axis represent TCWV in $kg/m^2$ **(a)** and TCC in % **(b)**. The X-axis indicates the time.



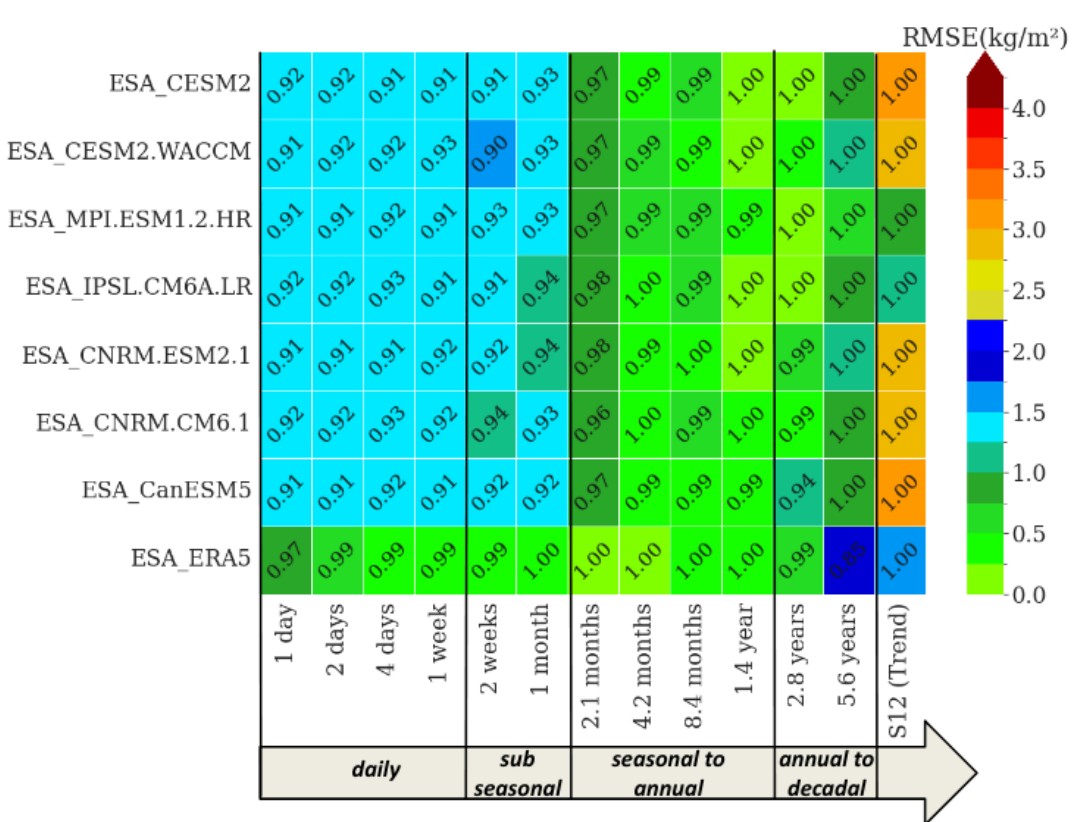





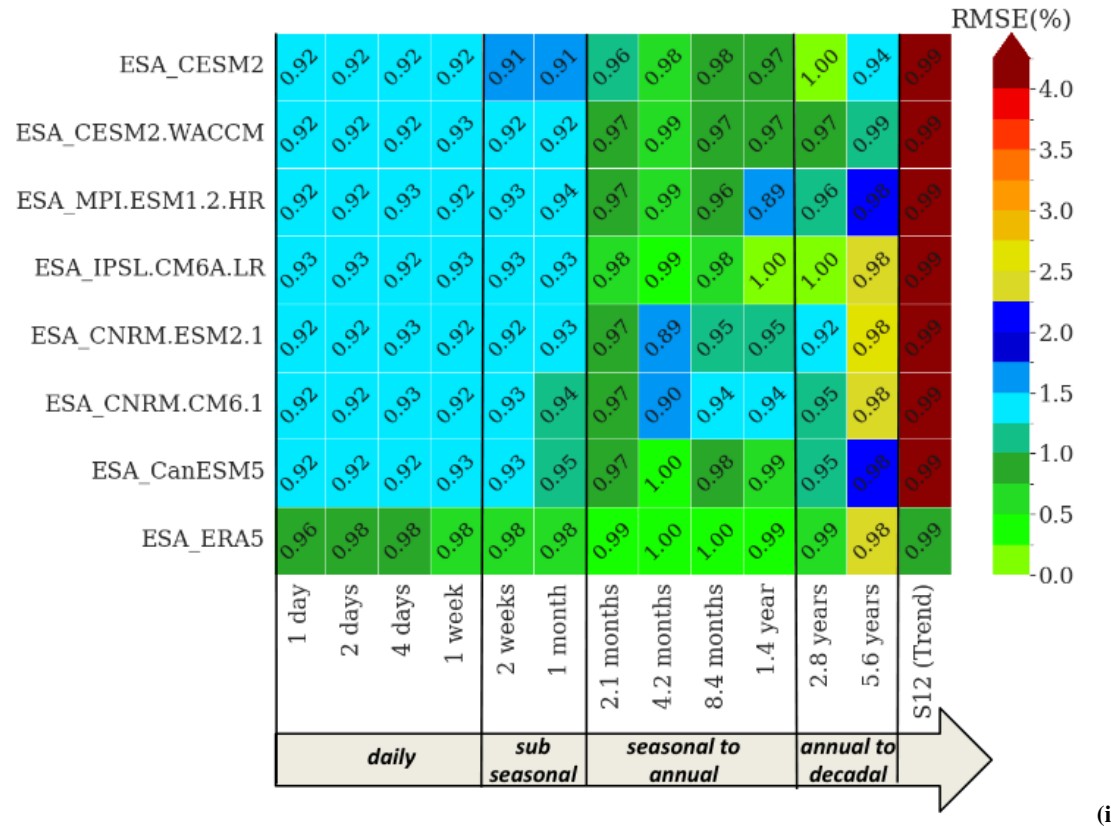

**Figure 7.** Evaluation of AMIP6 models and ERA5 with respect to ESA at different frequency for: **(i)** TCWV and **(ii)** TCC over the tropical oceans during the observed period. The Y-axis represents the different "**ESA_GCM$_j$**" and the X-axis the levels of decomposition or time scales. The numbers in the box indicate the correlation coefficient, while the color represents the RMSE.





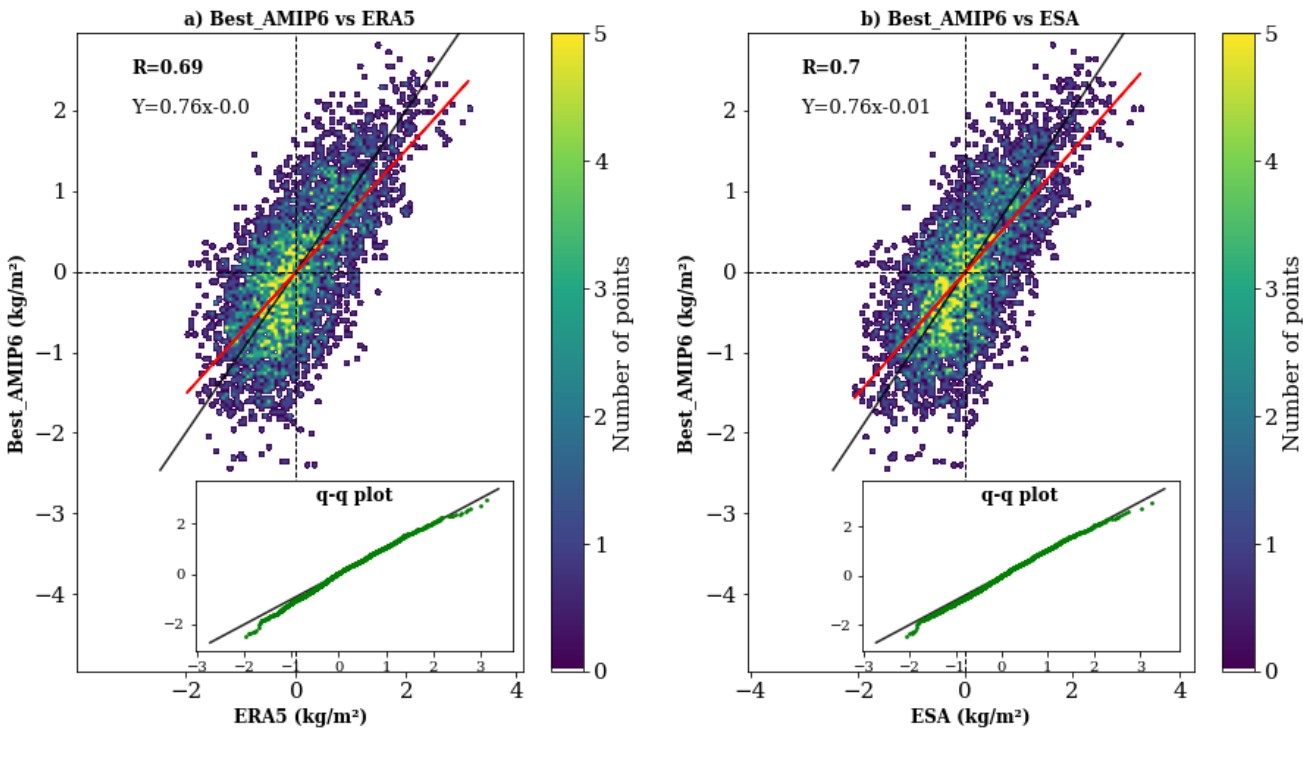

(i)



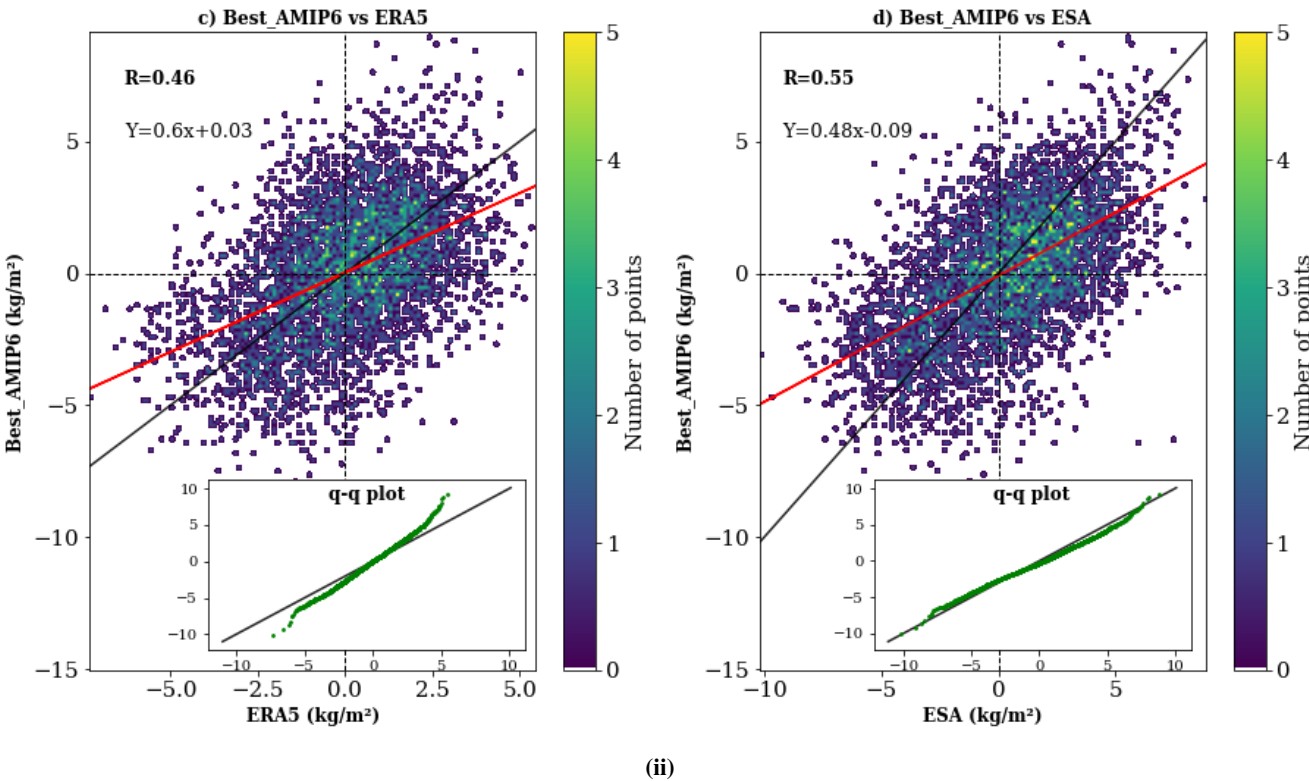

**(ii)**

**Figure 8.** Analysis of the correlation between the Best_CMIP6, ERA5 and ESA-CCI over the tropical oceans during the observed period for : **(i) TCWV** and **(ii) TCC**. The small integrated plots represent the quantile quantile plots. The black and red curves represent the identity line and the regression line respectively. The colorbar indicates the distribution of the points in the scatter plot. The Y-axis represents the anomalies of TCWV/TCC using the Best_AMIP6 and the X-axis, the anomalies of TCWV/TCC using ERA5 reanalysis or ESA-CCI. The anomalies are computed by removing the climatology in the evolution of signals. "R" indicates the pearson correlation coefficient, "Y" the equation of the regression line.







**Figure 9.** MRA decomposition of TCWV and TCC signals over the tropical oceans during the pre-ESA period. The decomposition is applied on 12 levels of resolution and the results are grouped in three time frequencies : **(a)** subseasonal, **(b)** seasonal to annual and **(c)** annual to decadal. The red, green and gray curves represent ESA-CCI, ERA5 and CMIP6. The Y-axis represent TCWV in $kg/m^2$ and TCC in %. The X-axis indicates the time.



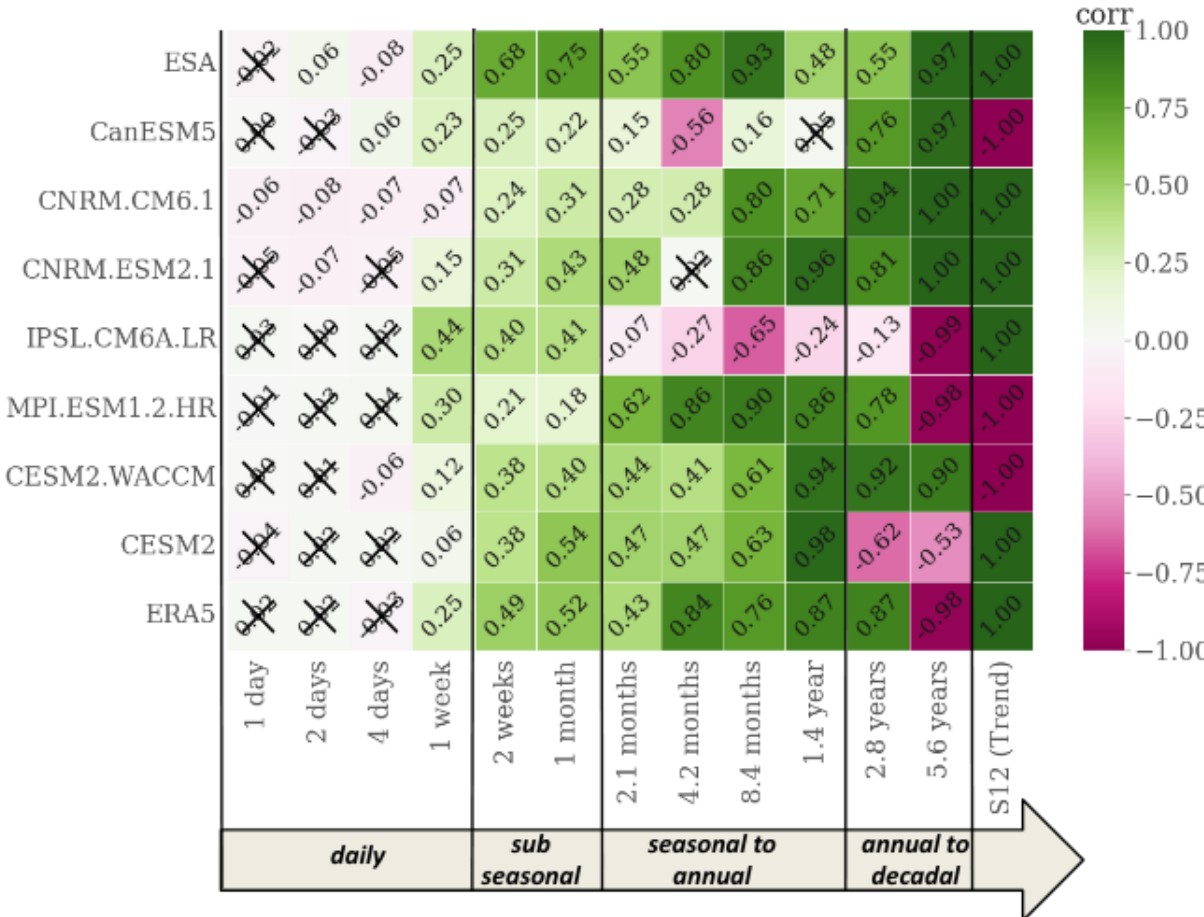

**Figure 10.** Evaluation of the co-variations between TCWV and TCC in ESA-CCI, AMIP6 and ERA5 at different frequency over the observed period. The Y-axis represents the different models and the X-axis the levels of decomposition or time scales. A significance test with a 99.9% confidence level was applied to the correlation between TCWV and TCC. The colorbar indicates the correlation between TCWV and TCC, while the crosses in the box indicate non-significant correlation values.