# Peer review of "General Circulation Models evaluation at different time scales over tropical region using ESA-CCI satellite data records: a case study of water vapour and cloud cover"

_EGUsphere, 2024_

## Author Comment (AC1)

**Comments Reviewer 1**

**Overview**

This study compared evolutions of the total column water vapor (TCWV) and total cloud cover (TCC) from AMIP6, ERA-5, and ESA-CCI over tropical ocean and land areas. The comparison was made for different time scales, using the multi-resolution analysis (MRA) decomposition method. TCWV signals from the AMIP6 are relatively well agreed with those of ESA observations, while larger differences are shown for the TCC signals. AMIP6 models also reproduced the variability of TCWV and TCC at seasonal to annual time scales. The authors also tried to attempt to select the best AMIP6 model group, based on the RMSE, and this group shows slight improvement compared to the entire AMIP6 group shown in this study.

This study was thoroughly performed, and the manuscript is well organized. The methodology used for the comparison for various time scales is solid, and the cited references are relevant. The display of figures, tables, and their corresponding captions is clear. As the authors mentioned, identifying the processes that are not well captured by models will give useful information to the satellite and climate modeling community. I recommend a minor revision before the publication in the Egusphere.

We thank the reviewer for taking the time to review our work. We have followed the reviewer's recommendations to improve the quality of the manuscript.

**General comments**

The TCWV and TCC from ESA are served as references throughout this study. The two observations are from the independent measurements. If there have been changes in the retrieval algorithm, sensor calibration, or operation of the satellite, there is a risk of the artifact in the long-term trends obtained from ESA products. I do not feel that the authors must investigate these factors specifically in this study. However, including relevant information in the methodology will give insight on the stability of the ESA products in analyzing long-term trends from the satellite products.

We thank the reviewer for raising this important point. We acknowledge that changes in retrieval algorithms, sensor calibration, or satellite operations can introduce uncertainties or artifacts in long-term climate data records. We added the following information in the main text at lines 88-90, in order to discuss that aspect of the data records :

<<ESA-CCI generally uses a variety of satellite instruments to retrieve Essential ECVs. To produce reliable climate data records for clouds, SST and water vapor, the stability and homogeneity of the satellite products are rigorously evaluated to ensure consistency across sensors and over time (Berry et al. (2018), Eliasson et al. (2019), Stengel et al. (2020), Trent et al.(2024)).>>

Eliasson, S., Karlsson, K. G., van Meijgaard, E., Meirink, J. F., Stengel, M., and Willén, U.: The Cloud_CCI simulator v1.0 for the Cloud_CCI climate data record and its application to a global and a regional climate model, Geosci. Model Dev., 12, 829-847, https://doi.org/10.5194/gmd-12-829-2019, 2019.

Stengel, M., Stapelberg, S., Sus, O., Finkensieper, S., Würzler, B., Philipp, D., Hollmann, R., Poulsen, C., Christensen, M., and McGarragh, G.: Cloud_CCI Advanced Very High Resolution Radiometer post meridiem (AVHRR-PM) data set version 3: 35-year climatology of global cloud and radiation properties, Earth Syst. Sci. Data, 12, 41–60, https://doi.org/10.5194/essd-12-41-2020, 2020.

Trent T., Schröder M., Ho SP, Beirle S, Bennartz R, and others., 2024. Evaluation of total column water vapour products from satellite observations and reanalyses within the GEWEX Water Vapor Assessment. *Atmospheric Chemistry and Physics.* European Geoscience Union. 16 9667–9695. https://doi.org/10.5194/acp-24-9667-2024

Berry, David I., Corlett, Gary K., Embury, Owen and Merchant, Christopher J. (2018) *Stability assessment of the (A)ATSR sea surface temperature climate dataset from the European Space Agency Climate Change Initiative.* Remote Sensing, 10 (1). 126. ISSN 2072-4292 doi: https://doi.org/10.3390/rs10010126

While the decomposition method shown in this study is very relevant, it could be improved if the physical reasons are discussed more. This discussion will serve as a key point for the modeling group to figure out why the models have disagreements with the observations.

We thank the reviewer for this insightful remark. We have added further discussion of the results presented in this study in the relevant sections, and the remaining points have been addressed in the corresponding comments :

**Section 3.1.1. lines 239-248** : << The weak correlation between TCC and SST can be attributed to several interconnected factors such as the regional and dynamic atmospheric circulation. Cloud formation depends not only on SST but also on atmospheric circulation patterns (e.g., Walker circulation, Hadley cells), moisture availability, and vertical motion (Stephens, 2005; Bony et al, 2015, Höjgard-Olsen et al, 2020). In some tropical regions, cloud variability is more strongly influenced by these

dynamics than by SST alone. TCC aggregates clouds of all types and altitudes, including low, mid, and high clouds, each responding differently to SST changes. For example, low-level stratocumulus clouds typically decrease with warming SSTs (negative correlation), while deep convective clouds tend to increase (positive correlation) (Klein and Hartmann, (1993); Bony and Dufresne (2005) and Stephens (2005)). These opposing effects can cancel out in TCC metrics, reducing the overall correlation. Taken together, these aspects suggest that the weak correlation between SST and TCC anomalies observed in this study does not necessarily indicate the absence of a relationship but rather highlights the complex, nonlinear nature of cloud–climate interactions in the tropics. >>

Stephens, G. L. (2005). Cloud feedbacks in the climate system: A critical review. *Journal of climate*, *18*(2), 237-273.

Bony, S., Stevens, B., Frierson, D. M., Jakob, C., Kageyama, M., Pincus, R., ... & Webb, M. J. (2015). Clouds, circulation and climate sensitivity. Nature Geoscience, 8(4), 261-268.

Klein, S. A., & Hartmann, D. L. (1993). The seasonal cycle of low stratiform clouds. *Journal of climate*, *6*(8), 1587-1606.

Höjgard-Olsen E., H Brogniez and H. Chepfer (2020). Observed evolution of the tropical atmospheric water cycle with sea surface temperature. *J. Climate, 33, 3449-3470, doi:10.1175/JCLI-D-19-0468.1*

Bony, S., & Dufresne, J. L. (2005). Marine boundary layer clouds at the heart of tropical cloud feedback uncertainties in climate models. *Geophysical Research Letters*, *32*(20).

**Section 3.1.3. lines 292-299** :

<<This contrast in TCWV trend over tropical lands compared to tropical oceans can be explained by a combination of physical factors. First, land surfaces warm more rapidly than oceans, leading to an increase in water vapor through the Clausius–Clapeyron relation (O'Gorman and Muller, 2010). Second, evapotranspiration over land adds more moisture to the atmosphere when soil moisture is sufficient, amplifying the trend (Betts, 2004; Seneviratne et al, 2010; Lorenz et al, 2016). In contrast, tropical oceans already maintain high humidity levels, so their relative changes over time are smaller (Sherwood et al, 2010). Additionally, changes in atmospheric circulation, such as monsoon shifts or enhanced moisture transport from ocean to land, may also contribute to this land–ocean contrast (Chou et al, 2001) >>

O'Gorman, P. A., & Muller, C. J. (2010). How closely do changes in surface and column water vapor follow Clausius–Clapeyronscaling in climate change simulations?. *Environmental Research Letters*, *5*(2), 025207.

Betts, A. K. (2004). Understanding hydrometeorology using global models. *Bulletin of the American Meteorological Society*, *85*(11), 1673-1688

Seneviratne, S. I., Corti, T., Davin, E. L., Hirschi, M., Jaeger, E. B., Lehner, I., ... & Teuling, A. J. (2010). Investigating soil moisture–climate interactions in a changing climate: A review. *Earth-Science Reviews*, *99*(3-4), 125-161

Lorenz, R., Argüeso, D., Donat, M. G., Pitman, A. J., van den Hurk, B., Berg, A., ... & Seneviratne, S. I. (2016). Influence of land‑atmosphere feedbacks on temperature and precipitation extremes in the GLACE‑CMIP5 ensemble. *Journal of Geophysical Research: Atmospheres*, *121*(2), 607-623

Chou, C., Neelin, J. D., & Su, H. (2001). Ocean‑atmosphere‑land feedbacks in an idealized monsoon. *Quarterly Journal of the Royal Meteorological Society*, *127*(576), 1869-1891.

**Specific comments**

The ESA-CCI data was used as a reference in this study. Therefore, it would be necessary to mention the uncertainty estimates of ESA-CCI TCWV and TCC for over ocean and land, around Table 1.

Thanks to the reviewer for this remark, we integrate such information in the table1 (see the last panel in the main document) and we added the following at lines 96-99

<< Due to its construction, the retrieval uncertainties of the ESA CCI TCWV data record depend on surface conditions (land / ocean)[Table1 (iii)]. Over oceans, TCWV estimated from microwave observations typically show lower uncertainties, ranging from 2 to 4 kg/m². In contrast, TCWV from near-infrared observations that are used over land are subject to greater uncertainty between 5 and 6 kg/m² or more, depending on the influence of surface reflectance and atmospheric conditions (Schröder et al. (2020)). For total cloud cover (TCC), uncertainties are generally lower over oceans (5–10% absolute cloud fraction), while retrievals over land are more uncertain (10–15%) because of surface heterogeneity and difficulties in detecting clouds over bright surfaces such as deserts or snow (Stengel et al., 2017, 2020)>>

Schröder, M., Hegglin, M., Ye, H., Falk, U.and Danne, O., Fischer, J., Laeng, A., Siddans, R., Sioris, C., Stiller, G., and Walker, K.: Water Vapour Climate Change Initiative (WV_cci) - CCI+ Phase 1, 2020

The ESA-CCI TCC parameter was obtained from multiple satellite sensors. Considering different spectral response functions, how consistent are the TCCs from those satellites? In other words, were any discontinuities or inconsistencies noted across satellite platforms? Including related references would be also useful.

We clarify this point in the main document by adding the following information in at lines 105 - 112 :

<<To ensure consistency across these various satellite platforms and minimize discontinuities, the ESA-CCI TCC team applied a rigorous inter-sensor harmonization process (Vukicevic et al. (2010) , Heidinger et al. (2014), Stengel et al. (2017)). This includes :

- spectral adjustment using radiative transfer simulations to correct for differences in spectral response functions;
- Cross-calibration and bias correction, often performed against a reference sensor or a stable long-term dataset;
- Homogenization techniques that reduce discontinuities and ensure temporal stability, particularly important for climate data records.

Although individual sensors have inherent differences in spectral response, no significant biases or discontinuities have been reported, and the dataset is considered reliable for climate-scale applications.>>

Vukicevic, T., Coddington, O., & Pilewskie, P. (2010). Characterizing the retrieval of cloud properties from optical remote sensing. *Journal of Geophysical Research: Atmospheres*, *115*(D20).

Heidinger, A. K., Foster, M. J., Walther, A., & Zhao, X. (2014). The pathfinder atmospheres–extended AVHRR climate dataset. *Bulletin of the American Meteorological Society*, *95*(6), 909-922.

Stengel, M., Stapelberg, S., Sus, O., Schlundt, C., Poulsen, C., Thomas, G., ... & Hollmann, R. (2017). Cloud property datasets retrieved from AVHRR, MODIS, AATSR and MERIS in the framework of the Cloud_cci project. *Earth System Science Data*, *9*(2), 881-904.

The ESA-CCI TCC was from polar orbit satellites, meaning that the TCC was sampled from specific local times for the region. In the longer scale of time series analysis, this may not impact the results, but further discussions might be necessary, particularly for the short time scale analysis.

This is actually an important point for the shortest time frequencies, we now discuss it in the main document by adding the information below at lines 112 - 122 :

<<It is important to note that ESA-CCI TCC is derived from polar-orbiting satellites (AVHRR-PM), which observe the Earth's surface at fixed local times typically 13:30 PM (e.g., NOAA-11, NOAA-14, NOAA-18, NOAA-19) (Stengel et al. (2017)). As a result, observations are usually limited to one pass per day. This temporal sampling constraint can significantly affect short-term analyses, such as the cloud diurnal cycle, potentially introducing temporal biases and misrepresenting cloud microphysics, particularly in convective regions (Pincus et al., 2012; King et al., 2013; Stubenrauch et al., 2013). However, the impact of this sampling bias decreases at monthly to seasonal time scales and becomes negligible at annual to decadal scales, as the diurnal sampling error tends to average out over longer periods. Since this study emphasizes the large-scale evolution of ESA-CCI TCC rather than local or short-term processes, the known temporal sampling limitations are unlikely to significantly affect our results.>>

Line 88: I guessed that WV_cci TCWV data are from the clear sky cases, according to the following paragraph about AMIP6 (line 113). If so, please mention here that WV_cci TCWV is from clear sky observations. Since the clear-sky sampling causes a dry-biased condition, it is worth mentioning here.

There is a misunderstanding here : while over lands the TCWV data is indeed for clear sky situations, over oceans the TCWV data is for nearly all sky conditions, since it is based on microwave measurements. We have clarified this point in the main text, by adding the following information at lines 99-101 :

<<The retrieval of WV_cci TCWV differs between land and ocean: over land, TCWV is retrieved under clear-sky conditions, whereas over ocean, it is retrieved under all-sky conditions (including cloudy, until convective precipitation occurs).>>

Lines 114 and 130: For AMIP6, a threshold of cloud mask of 50% was applied. In contrast, a threshold of cloud cover of 95% was applied to the ERA5 dataset. Please give a reason for this and discuss the impact of the threshold.

We agree with the reviewer that this point could be somewhat confusing. We clarify it by adding the following information at lines 156-159:

<<Following the ESA-CCI documentation (Falk et al. (2022)) and the methodology described by Sohn and Bennartz (2008) and applied by He et al. (2022), TCWV data are selected under different conditions for land and ocean surfaces. Over land (clear-sky conditions), only data with total cloud cover less than 95% and total column cloud liquid water below 0.005kg.m−2 are retained. Over the ocean, TCWV data are selected when total precipitation is below 0.001kg.m−2.s. >> This threshold was selected based on sensitivity analyses conducted by He et al. (2022) during their evaluation of TCWV representation in CMIP6 models. While assessing the impact of the threshold on our analysis could provide valuable insights, such an investigation lies beyond the scope of the present study.

Falk, U., Schröder, M., Brogniez, H., Eiras-Barca, J., Gimeno, L., He, J., Hubert, D., Lambert, J.-C., Preusker, R., Trent, T., & Hegglin, M. (2022). *Water Vapour Climate Change Initiative (WV_cci) – CCI+ Phase 1: Climate Assessment Report (CAR)* (Issue 3.1, ESA Report D5.1). European Space Agency (ESA) / ECSAT. https://climate.esa.int/media/documents/Water_Vapour_cci_D5.1_CAR_v3.1.pdf

Sohn, B. J., & Bennartz, R. (2008). Contribution of water vapor to observational estimates of longwave cloud radiative forcing. *Journal of Geophysical Research: Atmospheres*, *113*(D20).

He, J., Brogniez, H., & Picon, L. (2022). *Evaluation of tropical water vapour from CMIP6 global climate models using the ESA CCI Water Vapour climate data records*. *Atmospheric Chemistry and Physics*, 22, 12591–12606

Line 127: Isn't the resolution 0.05 degree?

The WV_cci dataset has a native resolution of 0.05°, while the ERA5 dataset is provided at a coarser resolution of 0.25°. Since line 127 specifically refers to the ERA5 dataset, the indication of a 0.25° resolution at this point in the text is correct and appropriate.

Line 170: The authors might have meant the power of a multiple of 2?

Thanks for this remark, we clarify in the text by adding : We changed "a multiple of power of 2 " by <<a multiple of a power of 2.>>

Figure 2: It seems that S12 shows the water vapor increase according to the global warming by following the Clausius-Clapeyron relation. Does the D12 signal represent the impact of ENSO signals? It would be great if the main contributor for the D12.

Thank to the reviewer for this remark, we provided a discussion by adding the following information in Section 3.1.1:

lines 227-228

<<The relationship between TCWV and SST is clearly evidenced by the similar trends in their evolution, as shown in Figure S1 of the supplementary material.>>

lines 230-237

<<The D12 component in Figure 2 represents the interannual variability of TCWV (up to approximately 6 years) and can partially reflect the influence of ENSO, a major driver of interannual climate variability in tropical regions. A negative correlation of -0.46 was found between the long-term evolution of ESA-CCI TCWV and sea surface temperatures (SST), indicating a relationship between the two variables. However, this also suggests that the long-term evolution of ESA-CCI TCWV is not only driven by ENSO. Other large-scale modes of variability, such as the Atlantic Multidecadal Oscillation (AMO) and the Pacific Decadal Oscillation (PDO), may also contribute to the observed signal. Further targeted analyses would be required to quantify the individual contributions of these climate modes. While such investigations would provide valuable insights, they are beyond the scope of the present study.>>

[Figure]

Figure S1 : Trend in the evolution of ESA-CCI TCWV and SST signals over the tropics during the period 2003-2014. TCWV in Kg/m² and SST in °C values are represented over the Y-axis while the X-axis indicates the date.

Line 216: AMIP6 models produce a revolution of TCWV close to the ESA-CCI, in terms of amplitude and phase, for the seasonal to annual time scales. In contrast, TCC from the AMIP6 models are quite different from ESA-CCI in terms of the phase and amplitude. The low-level cloud amounts are strongly tied to the SST anomalies. Therefore, I am wondering if the differences shown in TCC anomalies are mainly related to the mid- and high-level cloud evolution. A similar discussion is applied to the annual to decadal time scale. TCWV signals are relatively well agreed upon across ESA, ERA5, and AMIP6. However, larger differences are shown for the TCC signals (Fig. 5c).

I noticed that a short discussion is given in line 276 about the possible reasons for the larger differences in the TCC signals, but it would be great if the authors could elaborate further discussions or could include more references.

Thanks to the reviewer for this remark. We now elaborate more discussion on the differences in the representation of TCWV and TCC in the models. We add the following in the main text at lines 262-272:

<<This may result from the complexity of representing cloud microphysics and dynamics in large grid cells, as is the case with GCM. Cloud cover varies at small-scale and involves multiple cloud types, requiring subgrid parameterizations, which introduce uncertainty in the models (Stephens et al. (2005), Tompkins et al.(2005), Bony et al. (2015) Siebesma and Seifert.(2020), zelinka et al. (2020)). Low-level clouds, which are closely tied to surface conditions such as SST and near-surface stability, tend to be better represented, as these variables are often prescribed in AMIP simulations. However, mid- and high-level clouds, which are influenced by large-scale dynamics, vertical velocity fields, and convective processes, remain challenging to simulate accurately. These clouds contribute significantly to the overall TCC but are strongly dependent on the model's parameterizations of convection, microphysics, and cloud overlap assumptions. Further targeted diagnostics such as separating cloud layers, examining cloud types, and using complementary datasets like CALIPSO or CloudSat would be necessary to attribute the sources of disagreement and improve cloud representation in climate models.>>